# Molecular Evidence on the Inhibitory Potential of Metformin against Chlorpyrifos-Induced Neurotoxicity

**DOI:** 10.3390/toxics10040197

**Published:** 2022-04-18

**Authors:** Marzieh Daniali, Maryam Baeeri, Ramtin Farhadi, Mahdi Gholami, Shokoufeh Hassani, Mona Navaei-Nigjeh, Mahban Rahimifard, Mohammad Abdollahi

**Affiliations:** 1Department of Toxicology and Pharmacology, Faculty of Pharmacy, Tehran University of Medical Sciences, Tehran 11369, Iran; marziehdaniali75@gmail.com (M.D.); ramtinfarhadi@gmail.com (R.F.); m_gholami2068@yahoo.com (M.G.); shokoufehasani@gmail.com (S.H.); mahban.rahimifard@gmail.com (M.R.); 2Toxicology and Diseases Specialty Group, Pharmaceutical Sciences Research Center (PSRC), The Institute of Pharmaceutical Sciences (TIPS), Tehran University of Medical Sciences, Tehran 11369, Iran; mnavaei@sina.tums.ac.ir; 3Department of Pharmaceutical Biomaterials, Medical Biomaterials Research Center, Faculty of Pharmacy, Tehran University of Medical Sciences (TUMS), Tehran 11369, Iran

**Keywords:** brain, chlorpyrifos, *Galega officinalis*, metformin, neurotoxicity

## Abstract

Chlorpyrifos (CPF) is an organophosphorus (OP) pesticide, resulting in various health complications as the result of ingestion, inhalation, or skin absorption, and leads to DNA damage and increased oxidative stress. Metformin, derived from *Galega* *officinalis*, is reported to have anti-inflammatory and anti-apoptotic properties; thus, this study aimed to investigate the beneficial role of metformin in neurotoxicity induced by sub-acute exposure to CPF in Wistar rats. In this study, animals were divided into nine groups and were treated with different combinations of metformin and CPF. Following the 28 days of CPF and metformin administration, brain tissues were separated. The levels of inflammatory biomarkers such as tumor necrosis factor alpha (TNFα) and interleukin 1β (IL-1β), as well as the expression of 5HT1 and 5HT2 genes, were analyzed. Moreover, the levels of malondialdehyde (MDA), reactive oxygen species (ROS), and the ADP/ATP ratio, in addition to the activity of acetylcholinesterase (AChE) and superoxide dismutase (SOD), were tested through in vitro experiments. This study demonstrated the potential role of metformin in alleviating the mentioned biomarkers, which can be altered negatively as a result of CPF toxicity. Moreover, metformin showed protective potential in modulating inflammation, as well as oxidative stress, the expression of genes, and histological analysis, in a concentration-dependent manner.

## 1. Introduction

Chlorpyrifos (CPF) is an organophosphorus (OP) pesticide that has been heavily used and demonstrated various health complications in humans. CPF was introduced in 1965 as the most widely used pesticide in agriculture and non-agriculture environments [1]. According to the United States Environmental Protection Agency (EPA), elderly Americans consume CPF at 0.009 μg/kg daily through food and water consumption. However, ingestion, inhalation, and skin absorption are among the possible exposure routes of this pesticide [1,2]. The toxicity of CPF depends widely on the dose and duration of exposure; however, due to the half-life of CPF in water and food residue (as the main route of exposure for humans), many studies have focused on its subchronic exposures [1,2]. Different studies demonstrated the molecular mechanism of toxicity of CPF as DNA damage. Moreover, some studies reported CPF as a pesticide affecting the functionality and activity of cellular enzymes [3,4,5]. This pesticide is also reported to affect the regulation of serotonin receptors (5HT1 and 5HT2), which mediate hyperpolarization and the reduction of the firing rate of postsynaptic neurons [6,7].

Metformin is a biguanide drug originally developed from *Galega officinalis*, commonly known as goat’s rue or French lilac [8]. Since G. officinalis was found to contain guanidine and was associated with glucose-lowering potential, Jean Sterne initiated his studies of metformin as a glucose-lowering medication in the 1950s, and metformin was introduced as an antidiabetic medication in France and the United States in 1957 and 1995, respectively [9,10]. Nowadays, metformin is used as the first-line therapy in diabetes type 2 (T2D). Although the exact mechanism of metformin is not fully understood, suggested underlying mechanisms can be divided into two groups of adenosine 3′,5′-monophosphate (AMP)-dependent, and AMP-independent pathways [11]. Metformin is reported to have anti-inflammatory, anti-apoptotic, and anti-oxidative roles in different tissues [12,13,14,15,16]. Various studies also reported the beneficial impacts of metformin on diseases associated with the central nervous system (CNS), including reducing the risk of ischemic stroke and improvements in short-term neurological functions in traumatic brain injury [17,18]. Several in vivo studies noticed that chronic treatment with metformin (500 mg/kg for 30 days) reduced acetylcholinesterase (AChE) activity and levels of oxidative stress biomarkers [19,20]. Moreover, the amount of butyrylcholinesterase, which is involved in the prevalence of Alzheimer’s disease, was alleviated through the administration of metformin [21]. Thus, the focus of some recent studies was to investigate the role of metformin in Alzheimer’s disease, amnestic mild cognitive implications, Huntington’s, and Parkinson’s diseases [22,23,24,25].

Due to the lack of investigation on the role of metformin in CPF’s induced neurotoxicity, in the present study, the authors aimed to determine the beneficial role of metformin in attenuating the neurotoxicity symptoms of CPF. Biological, molecular, and analytical assays were performed after the administration of metformin in Wistar rats that were exposed to CPF for 28 days (sub-acute exposure).

## 2. Materials and Methods

### 2.1. Chemicals

The chemicals used in this study, including metformin and CPF, were purchased from Sigma-Aldrich^®^ (Munich, Germany). In addition, experimental kits for RNA extraction, cDNA synthesis, rat IL-1β and TNFα enzyme-linked immunosorbent assay (ELISA), and SOD activity were obtained from Sacace^®^ (Como, Italy), Thermo Scientific^®^ (Waltham, MA, USA), Diaclone^®^ (Besançon, France), and Teb Pazhouhan Razi^®^ (Tehran, Iran), respectively.

### 2.2. Animals

Healthy, adult, male Wistar rats weighing approximately 160 g were selected and kept in the animal house of the School of Pharmacy, Tehran University of Medical Sciences (TUMS). Wistar rats were adapted to the laboratory conditions two weeks before initiating the in vivo step. Animals were kept in an environment with a temperature of 25 ± 1 °C, humidity of 50–55%, and a 12 h light and dark cycle.

Moreover, all steps of this study were performed according to the regulations regarding animal experiments and it received ethical approval from the National Institute for Medical Research Development (NIMAD), on 18 November 2020, with the approval code of IR.NIMAD.REC.1399.257.

### 2.3. Study Design

In this study, animals were divided into nine groups of 6 Wistar rats. Proper concentrations of dissolved metformin in normal saline (NS) and CPF in corn oil were administered to the animals through intraperitoneal (IP) injection and oral gavage, respectively [1,2,3,4]. Administration of toxin and drug was performed in a sub-acute period (for 28 days), according to the following plan:Group 1 (Control corn oil): receiving oral gavage of corn oil;Group 2 (Control NS): receiving IP injection of NS;Group 3 (CPF): receiving oral gavage of 7.5 mg/kg (1/20 LD50) CPF;Group 4 (Met-30): receiving IP injection of 30 mg/kg/day metformin;Group 5 (Met-60): receiving IP injection of 60 mg/kg/day metformin;Group 6 (Met-120): receiving IP injection of 120 mg/kg/day metformin;Group 7 (CPF + Met-30): receiving oral gavage of 7.5 mg/kg CPF and IP injection of 30 mg/kg/day metformin;Group 8 (CPF + Met-60): receiving oral gavage of 7.5 mg/kg CPF and IP injection of 60 mg/kg/day metformin;Group 9 (CPF + Met-120): receiving oral gavage of 7.5 mg/kg CPF and IP injection of 120 mg/kg/day metformin.

After 28 days of CPF and metformin administration to the rats, ketamine and xylazine were injected into the rats at 100 mg/kg and 10 mg/kg, respectively. Four of the separated brain tissues of each group were frozen at −80 °C for further biochemical experiments, and 2 of the brain tissues were kept in 10 mL of 10% formalin, following washing with phosphate buffer two times (pH = 7.4).

### 2.4. Oxidative Stress Markers

First, 0.1 g samples of the cortex of the brain tissue were homogenized with 1 mL of phosphate buffer (PBS) and centrifuged for 5 min at 3000× *g*. The supernatant of the samples was collected for measuring oxidative stress biomarkers.

#### 2.4.1. Determination of ROS Level

ROS is produced through electron transport in mitochondria and is indicative of the free radicals associated with oxygen, leading to damages in cellular function [26]. First, 25 µL of the supernatant of the homogenized brain tissues was added to 81 µL of assay buffer and 5 µL of dichlorodifluorcein diacetate (DCFH-DA). Following 15 min of incubation at 37 °C, the fluorometric absorbance was measured for 60 min at the wavelength of 485 nm.

Moreover, to normalize the ROS level, protein level measurement was performed according to the Bradford Protein Assay (BPA). Here, 100 µL Bradford reagent was added to 10 µL of sample and was kept for 30 min in a dark place. The amount of protein was investigated by a spectrophotometer at the wavelength of 595 nm [27].

#### 2.4.2. Determination of MDA Level

The MDA level in brain tissue indicates the peroxidation of lipids and oxidative stress. First, 150 mL thiobarbituric acid of 1% w/v for was added to 600 mL supernatant. After placing the samples in boiling water for 15 min, 400 mL of n-butanol was added, and the level of MDA was investigated at the wavelength of 532 nm [28].

### 2.5. Determination of SOD Activity

First, 0.1 g samples of the cortex of the brain tissue were homogenized with 1 mL of KCl (150 mM) and centrifuged for 5 min at 3000× *g*. The supernatant of the samples was collected for the SOD activity assay. According to the kit manufacturer’s protocol, the activity of SOD was measured using a rat-specific enzyme-linked immunoassay (ELISA) kit.

### 2.6. Determination of Inflammatory Cytokine Levels (TNFα and IL-1β)

First, 0.1 g samples of the cortex of the brain tissue were homogenized with 1 mL of phosphate buffer and centrifuged for 15 min at 3000× *g*. The supernatant of the samples was collected for measuring inflammatory cytokines. Measurement of inflammatory biomarkers, including TNFα and IL-1β, was performed using rat-specific ELISA kits, according to the kit manufacturer’s protocol [29].

### 2.7. Determination of AChE Inhibition

First, 0.1 g samples of the cortex of the brain tissue were homogenized with 1 mL of phosphate buffer and centrifuged for 15 min at 3000× *g*. To determine the level of AChE inhibition, 10 μL of the homogenized samples were added to 3 mL of 5, 5′-dithiobis-(2-nitrobenzoic acid) (DTNB) solution (25 mM DTNB in 75 mM phosphate buffer). Following the addition of 10 μL of 3 mM acetylcholine iodide, a two-fold spectrophotometer was used to measure the absorbance change at the wavelength of 412 nm [30].

### 2.8. Gene Expression

The expression of specific genes, including 5HT1 and 5HT2, which are associated with the expression of serotonin receptors, was investigated by the real-time polymerase chain reaction (PCR) technique. In the first step, total RNA was extracted according to the Sacace^®^ RNA extraction kit protocol and then its concentration was measured by nanodrop. Complementary DNA (cDNA) was formed using a reverse transcription Thermo Scientific^®^ cDNA synthesis kit. In this study, the β-actin gene has been used as the housekeeping gene to study the expression of 5HT1 and 5HT2 genes. Finally, the double delta analysis was used to assess the expression of the mentioned genes. The sequences of the primers used in the real-time PCR step are listed in Table 1.

### 2.9. Determination of ADP/ATP Ratio

First, 0.1 g samples of the cortex of the brain tissue were homogenized in 1 mL of 6% cold trichloroacetic acid (TCA) solution. Following the centrifuging of the samples at 16,000× *g* for 10 min, the pH of the supernatants was neutralized using 0.5 M KOH solution. The neutral solutions were injected into a high-performance liquid chromatography (HPLC) instrument, and the ADP/ATP ratio was calculated and normalized according to the standard curves of ADP and ATP [31,32].

### 2.10. Histopathological Studies

The samples kept in formalin were embedded in paraffin, and 5 μm sections were prepared for staining with hematoxylin and eosin (H&E). Using an Olympus BX51 light microscope, histological slides from the cortexes of the brains were evaluated, and the changes in tissue sections, such as inflammatory responses, necrosis, hemorrhage, etc., were reported [33].

### 2.11. Statistical Analysis

Results of this study were presented as mean ± standard error of means (SEM). One-way analysis of variance (ANOVA) and Tukey’s multi-comparison tests were performed in GraphPad Prism, version 9.3.0. The significance of the changes was reported and set at *p* < 0.05.

## 3. Results

### 3.1. Oxidative Stress Biomarkers

#### 3.1.1. ROS

Results of the ROS test, summarized in Figure 1A, demonstrated that the administration of CPF to the animals led to a significant increase in the ROS level (*p*-value < 0.001). However, metformin in all three concentrations did not change the brain tissue’s ROS level, which indicates the safety of this medication. Despite the safety of metformin, the Met-30, Met-60, and Met-120 groups showed lower ROS markers in brain tissue in a concentration-dependent manner, and the Met-120 group resulted in the formation of the lowest level of ROS. Moreover, the groups receiving CPF and metformin simultaneously showed a concentration-dependent decrease in the ROS level of the brain tissue. However, the level of ROS in the CPF + Met-30 and CPF + Met-60 groups was significantly higher than in the control groups (*p*-value < 0.001 for both), the CPF + Met-120 group resulted in the most significant decline in ROS level in comparison with the CPF group (*p*-value < 0.001), and the lack of significant difference in ROS levels in comparison with the control groups demonstrated the beneficial role of metformin in the modulation of ROS markers in CPF-induced neurotoxicity.

#### 3.1.2. MDA

In the MDA test, which is provided in Figure 1B, the CPF group showed a significant increase in this marker in the brain tissue (*p*-value < 0.001). However, the lack of significant changes in the MDA marker in the groups receiving different concentrations of metformin indicated the safety of metformin in the formation of the MDA marker in brain tissue. Simultaneous administration of CPF and metformin to the animals resulted in fewer MDA biomarkers. In other words, the CPF + Met-30, CPF + Met-60, and CPF + Met-120 groups showed significantly decreased levels of MDA when compared to the CPF group, in a concentration-dependent manner (*p*-value < 0.05, *p*-value < 0.001, and *p*-value < 0.001, respectively). Results of the MDA test also demonstrated that the simultaneous administration of higher concentrations of metformin (60 mg/kg and 120 mg/kg) with CPF did not yield significant differences from the control groups, indicating its beneficial potential in the modulation of the MDA marker in CPF-induced neurotoxicity.

### 3.2. Inflammatory Cytokines

#### 3.2.1. TNFα

According to the results of the TNFα test, the CPF group showed significantly increased levels of the TNFα inflammatory cytokine compared to the control groups (*p*-value < 0.001). In contrast, the administration of metformin in 30 mg/kg, 60 mg/kg, and 120 mg/kg concentrations did not cause any changes in the level of the TNFα inflammatory cytokine. Results also demonstrated that the concentration-dependent administration of metformin with CPF could decrease the level of the TNFα cytokine compared to the CPF group. Although the level of the TNFα marker in the CPF + Met-30 and CPF + Met-60 groups did not show a significant difference in comparison with the CPF group but showed a significant difference from the control groups (*p*-value < 0.001 and *p*-value < 0.005, respectively), the CPF + Met-120 group showed a significant decrease in TNFα inflammatory cytokine levels in comparison with the CPF group (*p*-value < 0.05). Figure 2A summarizes the results of the TNFα cytokine test.

#### 3.2.2. IL-1β

Results of the IL-1β test confirmed the beneficial role of metformin in the modulation of inflammatory cytokines in CPF-induced neurotoxicity, which is demonstrated in Figure 2B. According to the results, the administration of CPF to the animals resulted in a significant increase in IL-1β inflammation cytokine levels. Moreover, metformin did not change the level of the IL-1β biomarker in the brain tissue at any of the three concentrations. However, the administration of metformin in the groups receiving CPF could alleviate the toxicity and IL-1β formation associated with CPF’s administration in a concentration-dependent manner. The co-administration of CPF and metformin in the CPF + Met-30, CPF + Met-60, and CPF + Met-120 groups demonstrated a decrease in the IL-1 marker level compared to the CPF group; however, the reductions in the IL-1β cytokine in the CPF + Met-60 and CPF + Met-120 groups were significant, with a *p*-value of <0.001 for both groups. Despite the modulatory impact of metformin in CPF-induced neurotoxicity, the CPF + Met-30, CPF + Met-60, and CPF + Met-120 groups showed significant differences when compared to the control groups (*p*-value < 0.001, *p*-value < 0.001, and *p*-value < 0.005, respectively).

### 3.3. AChE Inhibition

The results of AChE inhibition are explored in Figure 3. Results demonstrated that the administration of CPF in the animals significantly increased AChE inhibition in the brain tissue compared to the control groups (*p*-value < 0.001), which demonstrates the accumulation of ACh in the synaptic cleft, which can over-stimulate the relevant receptors, resulting in neurotoxicity. However, in the groups receiving 30 mg/kg, 60 mg/kg, and 120 mg/kg of metformin, the inhibition of AChE did not show any changes compared to the control groups, which confirms metformin’s safety profile in AChE inhibition. Results also demonstrated that metformin has modulatory potential to prevent the inhibition of AChE. These results also showed significant reductions in AChE inhibition in the CPF + Met-30, CPF + Met-60, and CPF + Met-120 groups (*p*-value < 0.005, *p*-value < 0.001, and *p*-value < 0.001, respectively). Moreover, the CPF + Met-60 and CPF + Met-120 groups did not show significant differences compared to the control groups, indicating the concentration-dependent modulatory impacts of metformin.

### 3.4. SOD Activity

According to the results obtained from the SOD activity test, the treatment of Wistar rats with CPF significantly reduced SOD activity compared to the control groups (*p*-value < 0.001). Although metformin showed neutral impacts on SOD activity and the groups receiving metformin at 30 mg/kg, 60 mg/kg, and 120 mg/kg concentrations did not show changes in the level of SOD activity, metformin could improve SOD activity when administrated with CPF. The concentration-dependent modulatory effect of metformin has been demonstrated in Figure 4. Elevated activity of SOD in the CPF + Met-30, CPF + Met-60, and CPF + Met-120 groups showed that although the SOD activity of CPF + Met-30 had a significant difference from that of the control groups (0.005), the lack of significant differences in the SOD activity of the CPF + Met-60 and CPF + Met-120 groups confirmed the beneficial role of metformin in the neurotoxicity induced by CPF.

### 3.5. Serotonin Receptor Gene Expression

#### 3.5.1. *5HT1*

According to the real-time PCR results, the expression of 5HT1 in the brain tissue of Wistar rats treated with CPF was significantly elevated compared to the control groups (*p*-value < 0.001). Moreover, the expression of the 5HT1 gene was not altered in the groups receiving metformin at 30 mg/kg, 60 mg/kg, and 120 mg/kg concentrations. Additionally, metformin showed a concentration-dependent, beneficial impact in alleviating the expression of this gene when administered simultaneously with CPF. Nonetheless, the CPF + Met-30, CPF + Met-60, and CPF + Met-120 groups showed significant reductions in 5HT1 expression when compared to the CPF group (*p*-value < 0.001), as well as significant increases in 5HT1 expression when compared to the control groups (*p*-value < 0.001). Figure 5A indicates the results of 5HT1 expression in the brain tissue through a real-time PCR test.

#### 3.5.2. *5HT2*

According to the real-time PCR results, the expression of 5HT2 in the brain tissue of Wistar rats treated with CPF was significantly elevated in comparison with the control groups (*p*-value < 0.001). Moreover, the expression of the 5HT2 gene was not altered in the groups receiving metformin at 30 mg/kg, 60 mg/kg, and 120 mg/kg concentrations. Additionally, metformin showed a concentration-dependent, beneficial impact in alleviating the expression of this gene when administered simultaneously with CPF. Nonetheless, the CPF + Met-30, CPF + Met-60, and CPF + Met-120 groups showed significant reductions in 5HT2 expression when compared to the CPF group (*p*-value < 0.005), as well as significant increases in 5HT2 expression when compared to the control groups (*p*-value < 0.001). Figure 5B indicates the results of 5HT2 expression in the brain tissue through a real-time PCR test.

### 3.6. ADP/ATP Ratio

The ADP/ATP ratio test results showed a significant increase in the ADP/ATP ratio in the group treated with CPF compared to the control group (*p*-value < 0.001). Moreover, the administration of lower metformin concentrations increased the ADP/ATP ratio in the brain tissue. In other words, although a significant difference was not observed in the Met-120 and control groups, the Met-30 and Met-60 groups showed an increased ADP/ATP ratio, with *p*-values of <0.005 and <0.05, respectively, which is associated with the AMP-dependent mechanism of metformin. Moreover, this concentration-dependent impact of metformin was observed in the groups exposed to CPF and metformin. These groups, including CPF + Met-30, CPF + Met-60, and CPF + Met-120, showed a significant increase in the ADP/ATP ratio compared to the control groups (*p*-value < 0.001 for all of the groups). However, in the CPF + Met-60 and CPF + Met-120 groups, the ADP/ATP ratio was significantly lower than in the CPF group (*p*-value < 0.001 for both groups). Nonetheless, the ADP/ATP ratio test results in the CPF + Met-120 group showed the lowest ratio among other groups receiving metformin and CPF simultaneously. Figure 6 demonstrates the results obtained from the ADP/ATP ratio test.

### 3.7. Histological Evaluation

Results of the histological analysis of the brain tissue are presented in Figure 7. In the control (corn oil) group (A), healthy tissue with glial cells (green arrows), perikaryon (red arrows), and axonal projections (brown arrows) was seen; likewise, no histological change was seen in the control (NS) group (B). Moreover, no histological alterations were seen in the Met-treated groups at the concentrations of 30 mg/kg, 60mg/kg, or 120 mg/kg (G, H, and I, respectively). The histological evaluations also showed that in the CPF group (C), tissue damage was observed, with vacuolated space and slight pyknosis (marked with yellow and gray arrows, respectively). However, some degree of vacuolization was observed in the CPF + Met-30 mg/kg group (D), as indicated with yellow arrows. Nevertheless, the CPF + Met-60 and CPF + Met-120 groups (E and F, respectively) were not reported to be histologically damaged, confirming the modulatory and beneficial role of metformin.

## 4. Discussion

Among OPs, CPF is linked to various neurological problems, and it is frequently used as an insecticide worldwide [34]. Several studies established different cellular pathways to explain the neurotoxicity of CPF [35,36,37,38]. CPF is associated with irreversible consequences in the brain tissue due to chronic exposure [5]. Despite the significant impairments of oxidative stress biomarkers, inflammatory cytokines, the activity of enzymes, and the expression of genes as the result of CPF exposure, metformin modulates the altered characteristics of the brain tissue [39]. Metformin, as a compound with natural botanic sources, is beneficial and preventive in various brain disorders [40]. Mechanisms impacting the formation of oxidative stress markers and the inflammatory cytokines are considered essential underlying mechanisms for the beneficial role of metformin [41].

In this study, metformin has been used to investigate its possible beneficial role in CPF-induced neurotoxicity. Oxidative stress biomarkers such as ROS and MDA, and inflammatory cytokines such as TNFα and IL-1β, were measured in this study. Results demonstrated that exposure to metformin alleviates and reduces the increased levels of the mentioned biomarkers associated with CPF. Moreover, the inhibition of AChE and the activity of SOD, which were increased and decreased, respectively, as a consequence of CPF treatment, were positively and significantly changed through metformin administration. Metformin can also impact the expression of the 5HT1 and 5HT2 genes and lower the expression levels to normal ones. Moreover, despite the histological injuries, including vacuolization and pyknosis, induced by CPF exposure, co-administration with the highest (120 mg/kg) concentrations of metformin did not demonstrate such damages in the brain’s histological sections. All of the tests performed in this study confirmed the role of metformin in modulating the neurotoxicity associated with CPF exposure and demonstrated the concentration-dependent impact of metformin.

Various studies investigated the influence of exposure to CPF, particularly neurotoxicity induced by CPF, which confirm the results obtained from our research. A study on Atlantic salmon demonstrated that CPF could affect protein degradation and lipid metabolism in the brain and liver, and also showed CPF’s impact on the disruption of encoding proteins involved in neuron function. This study found that CPF significantly altered the transcription of the genes involved in the neurological function of Atlantic salmon fish [35]. Moreover, another study on the neurotoxicity of CPF in mice reached the same results for AChE and SOD biomarkers. This study investigated the toxicity in the brain tissue and studied the abnormalities in AChE, SOD, catalase activity (CAT), glutathione peroxidase (GPX), and an increase in oxidative stress biomarkers. According to this study, CPF activates the formation of oxidative stress biomarkers and consequently alters significantly the activity of the mentioned enzymes [36].

AChE has been the focus of attention in various studies due to CPF-induced neurotoxicity. Although the exact mechanism of AChE alteration is not fully known, following the intracerebroventricular (ICV) injection of cytochrome P450 2B enzyme (CYP2B) inhibitors, the effect of the subcutaneous (SC) administration of CPF was assessed. Results of AChE neurochemical analysis showed that the CYP2B inhibitor attenuates the reduction in brain AChE. Thus, CYP2B is suggested as a factor involved in the neurotoxicity of CPF [37]. The expression of the genes involved in encoding AChE and monoamine oxidase A (MAO-A) has been examined. Results showed that the expression of the mentioned genes was reduced significantly when the animals were treated with CPF. This pesticide also significantly reduces the levels of neurotransmitters such as dopamine and serotonin and the activity of MAO-A, AChE, and sodium-potassium adenosine triphosphatase. The oxidative stress increase associated with exposure to CPF has been suggested to be relevant to a significant increase in MDA and nitric oxide (NO) markers [38]. Serotonin receptor assays have studied the impact of CPF on serotonin neurotransmitters. A study on an avian model showed that CPF directly increases the receptor binding of cerebrocortical 5HT2, demonstrating its upregulatory impacts on the expression of this serotonin receptor, and it also reduces the activity of presynaptic AChE in a concentration-dependent manner [42].

Furthermore, the role of CPF in 5HT signaling in noncholinergic neurotoxicity has been established. CPF is reported to alter the expected levels of the 5HT1A and 5HT2 receptors, in addition to 5HT transporters [7]. CPF is also suggested to modify the concentrations of pro-inflammatory and inflammatory biomarkers in the brain, plasma, and other tissues. Chronic exposure to CPF led to higher concentrations of TNFα, interleukin 6 (IL-6), and IL-1β in Wistar rats [43]. According to the literature, CPF impacts various cellular pathways in the brain tissue, and some of the most important mechanisms were discussed earlier. Due to the same cellular pathway of CPF and metformin, we benefited from metformin as the protective agent in our study.

The positive potential of metformin in neurotoxicity has been confirmed in several studies. Impaired mitochondrial oxidative metabolism as a result of insulin resistance is associated with cognitive decline. It is suggested to elevate ROS formation and consequently reduce mitochondrial ATP production. However, a recent study investigated metformin’s effect on mitochondrial proteins and mitochondrial fission, preventing ROS formation and inflammation [44]. Moreover, metformin showed beneficial roles in the cerebral ischemia of the brain by impacting mitochondrial dysregulation, oxidative stress, blood–brain barrier (BBB) breakdown, and inflammation. Various underlying mechanisms are suggested for the mentioned influences, including decreasing IL-6, TNFα, IL-1β, and intercellular adhesion molecule-1 (ICAM1) and consequent apoptosis prevention [45].

The neuroprotective role of metformin has also been studied in patients with acute stroke. The disrupted function of the brain as the result of impaired glucose control can be managed with the administration of metformin. This study suggested metformin’s role in controlling the blood glucose level, as well as altering the activated protein kinase (AMPK)/mammalian target of rapamycin (mTOR) signaling pathway and decreasing and increasing MDA and SOD biomarkers, respectively [46]. An animal study on streptozotocin (STZ)-induced diabetic rats showed that although brain injuries associated with diabetes are reported to reduce the activity of SOD to 65% and increase the MDA level to 59%, metformin has significant protective impacts against these injuries (*p*-value < 0.01) [47]. Another study on sepsis-induced brain injury showed that metformin ameliorates neuronal apoptosis by increasing the phosphorylation of protein kinase B (PKB) and activating phosphoinositide 3-kinase (PI3K)/Akt signaling [48].

## 5. Conclusions

The main conclusion to be drawn is that CPF-induced neurotoxicity is associated with increased levels of oxidative stress biomarkers as well as inflammatory cytokines. Moreover, CPF can impair the activity of SOD while increasing the expression of the genes relevant to serotonin receptors. CPF alters the ratio of ADP to ATP, and this pesticide can result in histological injuries in the brain tissue. Administration of metformin is reported to modulate the changes associated with the neurotoxicity of CPF, and metformin demonstrated its beneficial impacts in a concentration-dependent manner. Thus, this study suggests metformin as a protective agent against the neurotoxicity of CPF. Moreover, in future studies, other biomarkers and pathways can be studied with sub-acute and chronic exposure to CPF.

## Figures and Tables

**Figure 1 toxics-10-00197-f001:**
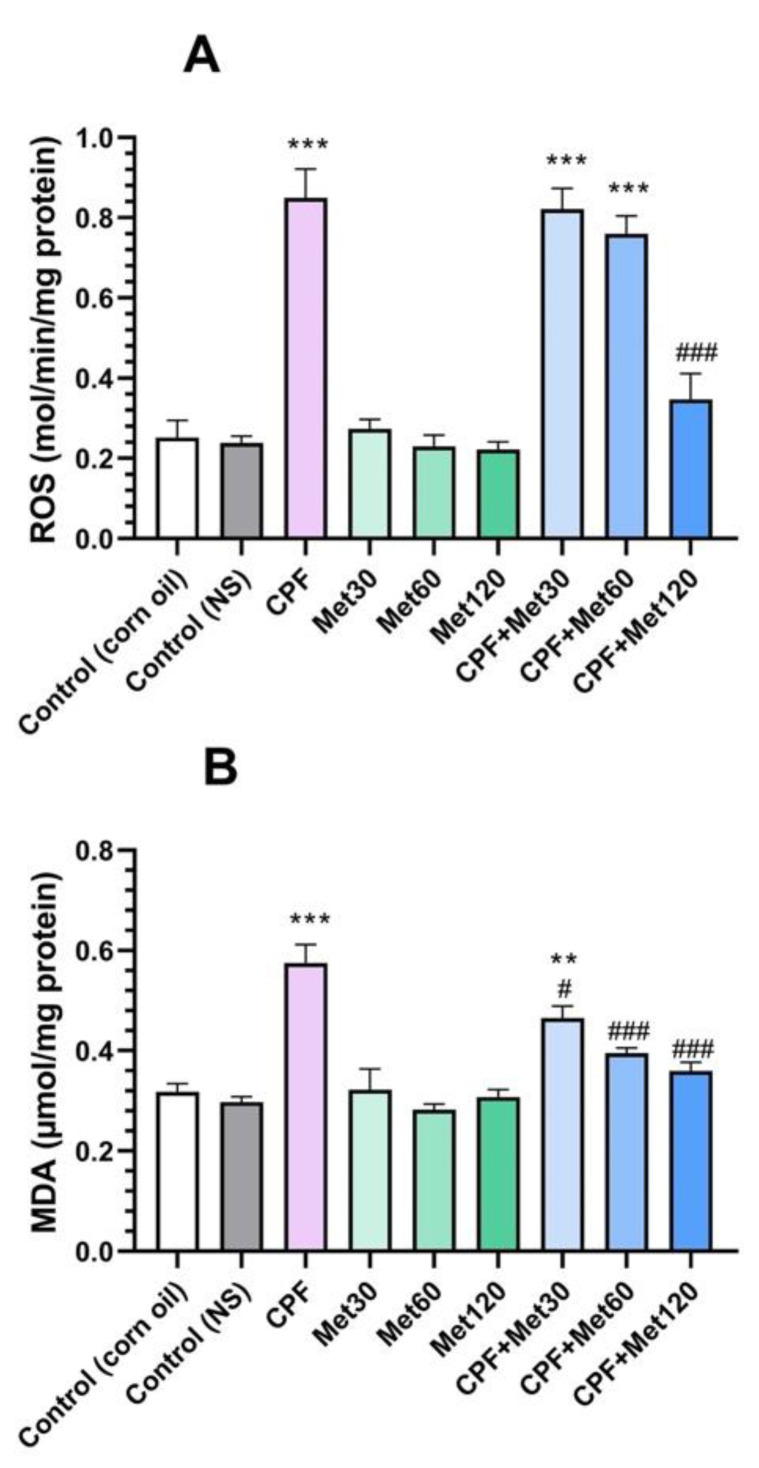
Results of oxidative stress biomarker tests on the brain tissue of 6 Wistar rats in each group. Data were obtained from 4 repeated measurements and are reported as the mean ± SEM. (**A**) Reactive oxygen species (ROS) assay. (**B**) Malondialdehyde (MDA) assay.** *p*-value < 0.005, *** *p*-value < 0.001; compared with the control groups. # *p*-value < 0.05 and ### *p*-value < 0.001; compared with chlorpyrifos (CPF) group.

**Figure 2 toxics-10-00197-f002:**
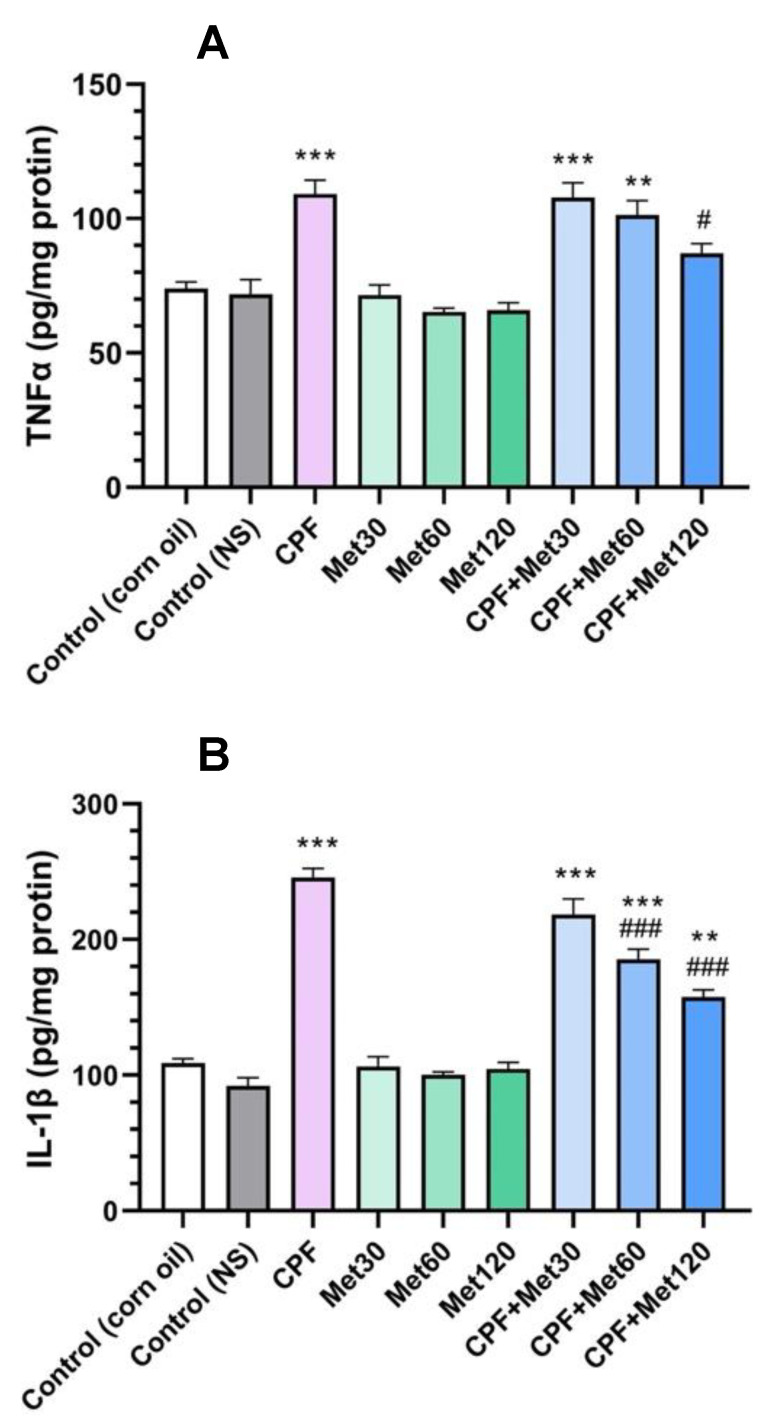
Results of inflammatory cytokine tests on the brain tissue of 6 Wistar rats in each group. Data were obtained from 4 repeated measurements and are reported as mean ± SEM. (**A**) Tumor necrosis factor alpha (TNFα) assay. (**B**) Interleukin 1 beta (IL-1β) assay. ** *p*-value < 0.005, *** *p*-value < 0.001; compared with the control groups. # *p*-value < 0.05 and ### *p*-value < 0.001; compared with chlorpyrifos (CPF) group.

**Figure 3 toxics-10-00197-f003:**
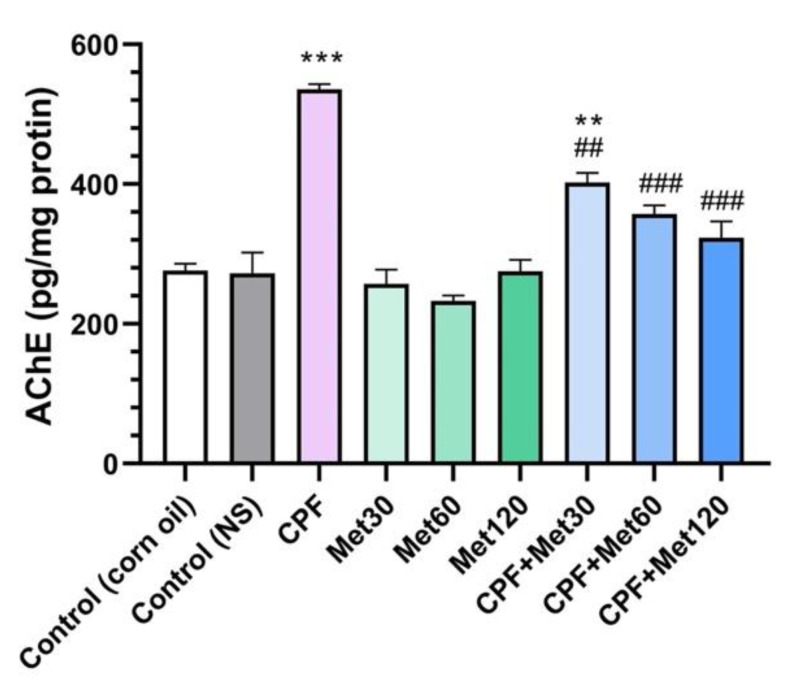
Results of acetylcholinesterase (AChE) inhibition test on the brain tissue of 6 Wistar rats in each group. Data were obtained from 4 repeated measurements and are reported as mean ± SEM. ** *p*-value < 0.005, *** *p*-value < 0.001; compared with the control groups. ## *p*-value < 0.005, ### *p*-value < 0.001; compared with chlorpyrifos (CPF) group.

**Figure 4 toxics-10-00197-f004:**
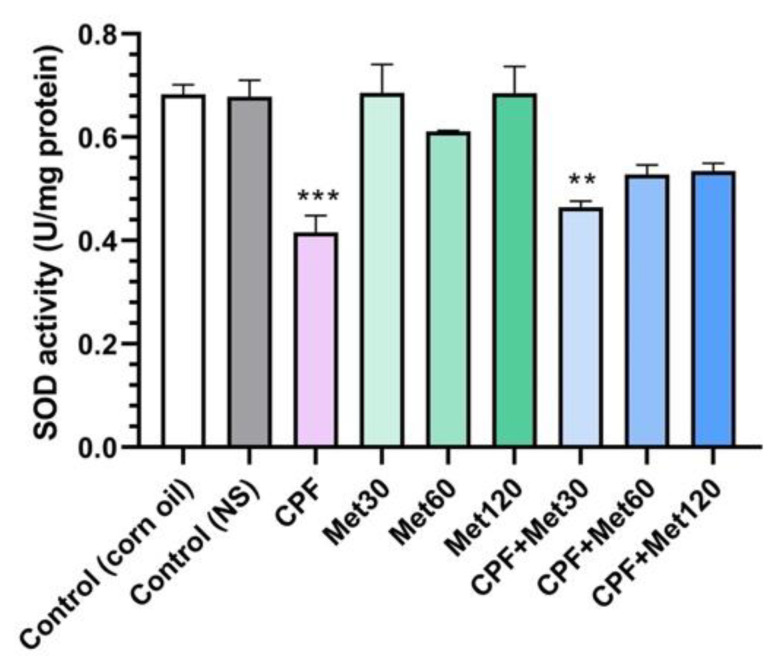
Superoxide dismutase (SOD) activity assay on the brain tissue of 6 Wistar rats in each group. Data were obtained from 4 repeated measurements and are reported as mean ± SEM. ** *p*-value < 0.005, *** *p*-value < 0.001; compared with the control groups.

**Figure 5 toxics-10-00197-f005:**
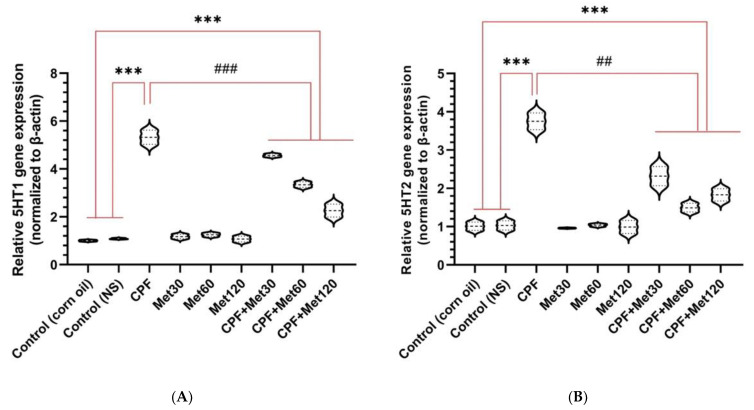
Expression of the genes associated with serotonin receptors (5HT1 and 5HT2, in (**A**,**B**), respectively) in the brain tissue of 6 Wistar rats in each group. Data were obtained from 4 repeated measurements and are reported as mean ± SEM. *** *p*-value < 0.001; compared with the control groups. ## *p*-value < 0.005 and ### *p*-value < 0.001; compared with chlorpyrifos (CPF) group.

**Figure 6 toxics-10-00197-f006:**
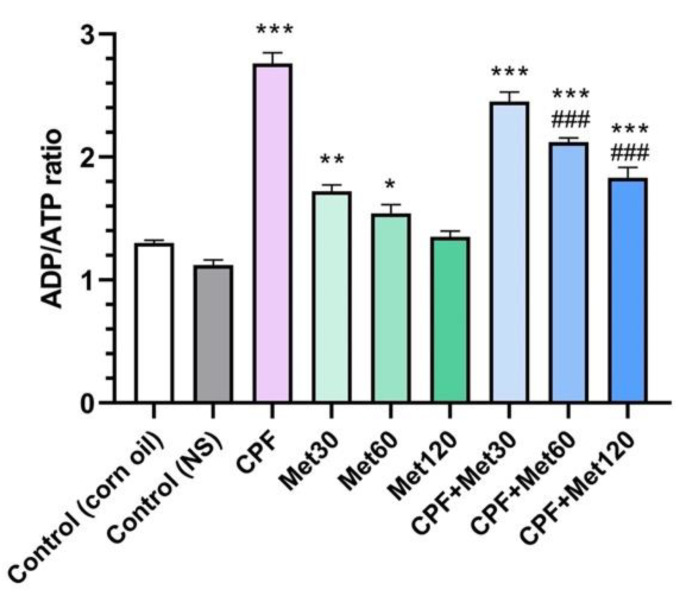
Results of ADP/ATP ratio assay on brain tissue of 6 Wistar rats in each group. Data were obtained from 4 repeated measurement and are reported as mean ± SEM. * *p*-value < 0.05, ** *p*-value < 0.005, *** *p*-value < 0.001; compared with the control groups. ### *p*-value < 0.001; compared with chlorpyrifos (CPF) group.

**Figure 7 toxics-10-00197-f007:**
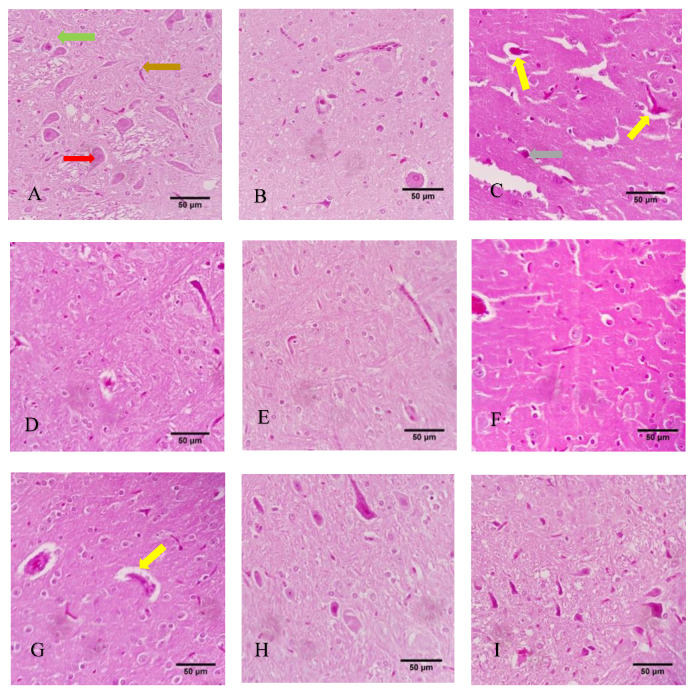
Histological sections of the cortex of the brain tissue of 6 Wistar rats in each group. Yellow, gray, green, red, and brown arrows indicate vacuolization, pyknosis, glial cells, perikaryon, and projections. (**A**) Control (corn oil), (**B**) Control (NS), (**C**) CPF, (**D**) Met-30, (**E**) Met-60, (**F**) Met-120, (**G**) CPF + Met-30, (**H**) CPF + Met-60, (**I**) CPF + Met-120.

**Table 1 toxics-10-00197-t001:** Sequences of the genes used in real-time PCR analysis.

Name	Symbol	Primer Sequence
Rattus norvegicus actin, beta (Actb)	β-actin	F: AGGGAAATCGTGCGTGACAT R: CCGATAGTGATGACCTGACC
Rattus norvegicus 5-hydroxytryptamine receptor 1A (Htr1a)	5HT1	F: GTCCACTTGTTGAGCACCTG R: ACGTGACCTTCAGCTACCAA
Rattus norvegicus 5-hydroxytryptamine receptor 2A (Htr2a)	5HT2	F: TAGTTTGGCTCGAGTGCTGA R: TCCATGCCAATCCCAGTCTT

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
