# Peer review of "Molecular Evidence on the Inhibitory Potential of Metformin against Chlorpyrifos-Induced Neurotoxicity"

_toxics, 2022, doi:10.3390/toxics10040197_

Round 1
Reviewer 1 Report
General comment:
The manuscript brings useful information about the inhibitory potential of metformin against chlorpyrifos (CPF)-induced neurotoxicity. The results are interesting, and the authors showed that exposure to metformin alleviates and reduces the increased levels of oxidative stress biomarkers (i.e., ROS and MDA), and inflammatory cytokines (i.e., TNF-α and IL-1β) biomarkers associated with CPF. The authors claimed that metformin treatment has the potential in alleviating the oxidative stress biomarkers and inflammatory cytokines which can be altered negatively as a result of CPF toxicity. Finally, the authors concluded that metformin showed protective potential in modulating inflammation as well as oxidative stress, expression of genes and histological analysis, in a concentration-dependent manner. The reviewer believed that the present study is interesting and potentially could contribute to the research field, however, there are only a few concerns that require to be addressed to clarity and improve the present version.
Specific comments:
Title
The title is informative and relevant to the major findings.
Abstract
In the abstract, the aim of the study is clearly mentioned, and major results are also properly presented.
Introduction
The research gap/question is not properly outlined. Please clearly mention the research gap and objectives of the study in this section.
Materials and Method
In general, well described. However, statistical analysis is not well explained, no information on the used software package and version, replication of individual experiments.
Results
The results section is well-organized and well explained. However, it would be better if the authors could compare their results with other previous reports
Discussion
No major issues.
Conclusion
The conclusion does not properly answer the aims of the study. Major limitations and opportunities to inform future research are not addressed properly.
Author Response
Dear Respected Reviewer,
Thank you so much for reviewing our work and allowing us to revise it. We have hopefully met all your comments and tried our best to edit the article (the changes are highlighted in yellow). We hope you will find our changes satisfactory and suitable for publication.
Point-by-point responses:
Introduction: The last paragraph of the introduction section has been changed to: Due to the lack of investigation on the role of metformin in CPF’s induced neurotoxicity, in the present study, the authors aimed to determine the beneficial role of metformin in attenuating the neurotoxicity symptoms of CPF. Biological, molecular, and analytical assays were performed after administration of metformin in Wistar rats that were exposed to CPF for 28 days (sub-acute exposure).
Materials and Method: The data on the used software was added to section 2.11. and replication of individual experiments was described in section 2.3.
Result: The comparison of the results of our study with the results of previous reports has been completely performed in the discussion section.
Conclusion: The conclusion section was changed in order to fulfill the comments of the reviewer.
Reviewer 2 Report
This manuscript described that metformin alleviated the neurotoxicity due to CPF-induced oxidative damage. To elucidate the mechanism, the authors measured inflammatory cytokines, levels of oxidative stress biomarkers, expression of genes associated with serotonin receptors, SOD activity and the ratio of ADP to ATP. The results shown were generally sound and supported the conclusions. However, there are some grammatical and typographical errors in the manuscript. English clear and should be edited with help of a native scientist or a commercially available English proofreader. In addition, there are some points which the authors should address as follows;
- Dis the authors comply with the regulations regarding animal experiments? Also, have their experiments been deliberated and approved by the Institutional Ethics Board? If so, please specify them in the text.
- Grouping rats was a bit complicated. Did the authors need two types of controls? Was it necessary to have 3 groups of metformin alone?
- Please specify the sample size of each experiment (probably maximum 6 or less) in the legend of each figure.
- In “Results”, 180 to 182 lines were no need. Please delete them.
- In Figure 1A, did the amounts of ROS in the CPF + MET30 group and the CPF + MET60 group significantly decrease compared to the amount of ROS in the CPF alone administration group?
- What can be learned by measuring AchE inhibition? Add it briefly in the AchE Inhibition section of Methods or Results.
- In Figure 4, please clarify whether the SOD activity in the CPF + MET combined groups was significantly increased compared to the SOD activity in the CPF alone administration group, or not.
- The notation in Figure 5 was confusing. Please express it like any other figures.
- Why did 30 and 60 mg / kg / day doses of metformin significantly increase the ADP / ATP ratio compared to that in controls? What changes does the ADP / ATP ratio correspond to?
Author Response
Dear Respected Reviewer,
Thank you so much for reviewing our work and allowing us to revise it. We have hopefully met all your comments and tried our best to edit the article (the changes are highlighted in yellow). We hope you will find our changes satisfactory and suitable for publication.
Point-by-point responses:
- The date regarding the ethical approval of animal experiments has been added to 2.2. section.
- Two types of control groups were used to make sure that either normal saline or corn oil cannot result in toxicity in brain tissue. Also, the concentration-dependent role of metformin can be confirmed when all the doses were investigated alone, as well as in combination with CPF. The other reason for using metformin in 3 different concentrations was to make sure that metformin, even in its highest dose (120 mg/kg) cannot alter significantly the biomarkers in the brain and result in neurotoxicity.
- The sample size of each experiment was added to the legend of each figure.
- This part was omitted.
- No, actually the amount of ROS in CPF (alone) and CPF+Met30 and CPF+Met60 were significantly higher than in the control groups (demonstrated with *) and CPF+Met120 showed that it has significantly lower levels of ROS in comparison with the CPF(alone) group (demonstrated with #).
- The data regarding the inhibition of AChE in neurotoxicity was added in section 3.3.
- The increased levels of SOD activity in CPF+ Met groups are not demonstrated through # signs, therefore, the SOD activity is not significantly altered through the addition of metformin in comparison with CPF group. However, because of the lack of * signs in CPF+Met 60 and CPF+Met 120, it can be obtained that the addition of metformin modulates the alteration of SOD in comparison with the control groups and the activity of SOD in the mentioned groups are closer (not significantly different) with the control groups.
- This part has been changed.
- As mentioned in the introduction section, the exact mechanism of metformin is not fully understood but it might be associated with AMP and ATP. Therefore, alteration of ADP/ATP ratio in Met groups is normal. However, the reason why Met 120 mg/kg did not result in the same alteration is suggested to be investigated in future studies.
Reviewer 3 Report
The study entitled “Molecular evidence on the inhibitory potential of metformin against chlorpyrifos-induced neurotoxicity” by Daniali, M. et al. examined the protective role of metformin in the organophosphorus pesticide treated rat brain by employing various neurotoxicity tests. Overall, this study was well designed, carefully executed with a set of appropriate experimental procedures, and results properly interpreted. This paper should be of interest to the journal’s targeted audience. However, the reviewer has some major concerns and a few minor concerns as outlined below.
Major concern:
(1) The nature of the study is rather descriptive. There is no solid mechanistic data supporting the findings.
(2) It is understandable why the oxidative stress markers and inflammatory cytokines are examined. But why the serotonin receptors (HT1 & HT2)? There is no rationale of why these two neurotransmitter receptors were selected and no explanation of how their induction by CPF can contribute to the neuronal toxicity. Furthermore, in the discussion, the authors cited a study which showed the opposite finding (i.e., “metformin decreases SOD levels to 65% and increases MDA levels to 59% [46]” lines 445-447 on page 14). The authors need to clarify such discrepancy and provide further discussion on this issue.
(3) Another discrepancy is on page 13, lines 418-419. In the study cited by the author, CPF reduces expression of 5HT1 but the authors presented opposite findings in the manuscript. This need to be clarified and explained as well.
Minor concerns:
There are places that appear to be confusing to the reviewer, which may likely be so to the readers, too. Such unclear writing makes the manuscript hard to understand and thus will benefit from a thorough edits to improve its readability.
Examples include:
(1) The last three paragraphs in the Discussion are confusing to the reviewer.
(2) Page 1, line 398-400, it is not clear how “early fetal mice” is defined by the author. How early, and at what fetal stage?
(3) Page 5, lines 180-182, it seems the author copy & paste the journal’s instruction here.
(4) Page 14, lines 472-474. Same issue as above
Author Response
Dear Respected Reviewer,
Thank you so much for reviewing our work and allowing us to revise it. We have hopefully met all your comments and tried our best to edit the article (the changes are highlighted in yellow). We hope you will find our changes satisfactory and suitable for publication.
Point-by-point responses:
Major concerns:
- According to the not-fully known mechanism of metformin, we tried to analyze different biomarkers and factors to study the role of metformin in CPF-induced toxicity. Also, the biomarkers which were investigated in this study can shed light on prospective molecular pathways which are important to be analyzed in future studies.
- Serotonin receptors are associated with glutamate-induced and NMDA-stimulated neurotoxicity and cGMP accumulation. Therefore, examination of the expression of 5HT1 and 5HT2 genes are important for neurotoxicity in cortical cells. Also, the part regarding the impact of diabetes on MDA and SOD has been changed.
- The part regarding the “Developmental neurotoxic effects of chlorpyrifos on acetylcholine and serotonin pathways in an avian model” study has been changed.
Minor concerns:
- These paragraphs have been rewritten.
- This part has been rewritten.
- This part has been omitted.
- This part has been omitted.
Round 2
Reviewer 2 Report
The manuscript has been well revised.
Reviewer 3 Report
Publish as is.